# Early decarbonisation of the European energy system pays off

Marta Victoria [1,2✉], Kun Zhu[1], Tom Brown [3], Gorm B. Andresen[1,2] & Martin Greiner [1,2]

For a given carbon budget over several decades, different transformation rates for the energy system yield starkly different results. Here we consider a budget of 33 GtCO₂ for the cumulative carbon dioxide emissions from the European electricity, heating, and transport sectors between 2020 and 2050, which represents Europe's contribution to the Paris Agreement. We have found that following an early and steady path in which emissions are strongly reduced in the first decade is more cost-effective than following a late and rapid path in which low initial reduction targets quickly deplete the carbon budget and require a sharp reduction later. We show that solar photovoltaic, onshore and offshore wind can become the cornerstone of a fully decarbonised energy system and that installation rates similar to historical maxima are required to achieve timely decarbonisation. Key to those results is a proper representation of existing balancing strategies through an open, hourly-resolved, networked model of the sector-coupled European energy system.

[1] Department of Engineering, Aarhus University, Inge Lehmanns Gade 10, 8000 Aarhus, Denmark. [2] iCLIMATE Interdisciplinary Centre for Climate Change, Aarhus University, Aarhus, Denmark. [3] Institute for Automation and Applied Informatics (IAI), Karlsruhe Institute of Technology (KIT), Forschungszentrum 449, 76344 Eggenstein-Leopoldshafen, Germany. ✉email: mvp@eng.au.dk

Achieving a climate-neutral European Union in 2050[1] requires meeting the milestones in between. Although carbon emissions will most likely sink by 20% in 2020 relative to 1990[2], it is unclear whether the 40% objective settled for 2030 will be met. The national energy plans for the coming decade submitted by member states do not add up the necessary reduction to meet the target[3], while in the context of a European Green Deal a more ambitious reduction of 55% is under discussion in 2020[4].

A remaining global carbon budget of 800 Gigatons (Gt) of $CO_2$ can be emitted from 2018 onwards to limit the anthropogenic warming to 1.75 °C relative to the preindustrial period with a probability of >66%[5]. This is compatible with holding the temperature increase well below 2 °C as stated in the Paris Agreement. Different sharing principles can be used to split the global carbon budget into regions and countries[6]. Subtracting the $CO_2$ emissions in 2018 and 2019, and considering an equal per-capita distribution translates into a quota of 48 $GtCO_2$ for Europe. An approach that took into account historical emissions would lead to more ambitious targets for Europe than other regions[7]. Assuming that sectoral distribution of emissions within Europe remains at present values, the carbon budget for the generation of electricity and provision of heating in the residential and services sectors accounts for ~21 $GtCO_2$,[8] and Supplementary Note 1. The budget increases to 33 $GtCO_2$ when the transport sector is included.

Electricity generation is expected to spearhead the transition spurred by the dramatic cost reduction of wind energy[9] and solar photovoltaics (PV)[10,11]. A vast body of literature shows that a power system based on wind, solar and hydro generation can supply hourly electricity demand in Europe as long as proper balancing is provided[12–15]. This can be done by reinforcing interconnections among neighbouring countries[16] to smooth renewable fluctuations by regional aggregation or through temporal balancing using local storage[17–19]. Moreover, coupling the power system with other sectors could provide additional flexibilities facilitating the system operation and simultaneously helping to abate emissions in those sectors[20–22].

$CO_2$ emissions from heating in the residential and services sectors show a more modest historical reduction trend compared to electricity generation (Fig. 1). Nordic countries have been particularly successful in reducing carbon emissions from the

heating sector by using sector-coupling strategies, Supplementary Figs. 2 and 3. Denmark, where more than half of the households are connected to district heating systems[23], has shifted the fuel used in Combined Heat and Power (CHP) units from coal to biomass and urban waste incineration[24]. Sweden encouraged a large-scale switch from electric resistance heaters to heat pumps[23] which are now supported by high $CO_2$ prices[25] and low electricity taxes.

Energy models assuming greenfield optimisation, that is, building the European energy system from scratch without considering current capacities, shows that sector-coupling decreases the system cost and reduces the need for extending transmission lines due to the additional local flexibility brought by the heating and transport sectors[21]. Sector-coupling allows large $CO_2$ reductions before large capacities of storage become necessary, providing more time to further develop storage technologies[19]. Greenfield optimisation is useful to investigate the optimal configuration of the fully decarbonised system, but it does not provide insights on how to transition towards it. Today's generation fleet and decisions taken in intermediate steps will shape the final configuration.

Transition paths for the European power system have been analysed using myopic optimisation, i.e., without full foresight over the investment horizon[26–29]. Myopic optimisation results in higher cumulative system cost than optimising the entire transition period with perfect foresight because the former leads to stranded investments[28,30]. However, the myopic approach is less sensitive to the assumed discount rate and can capture better short-sighted behaviour of political actors and investors[28,29].

Transition paths under stringent carbon budgets have been mainly investigated using Integrated Assessment Models (IAMs), which represent a broader approach including other sectors, globe, land and climate models[10,31–33]. However, the low temporal resolution and outdated cost assumptions for wind and solar PV[10,34] in IAMs could hinder the role that renewable technologies could play in decarbonising the energy sector.

In this work, we use an hourly resolved sector-coupled networked model of the European energy system and myopic optimisation in 5 years steps from 2020 to 2050 to investigate the impact of different $CO_2$ reduction paths with the same carbon budget. In every time step, the expansion of generation, storage and interconnection capacities in every country is allowed if it is cost-effective under the corresponding global emissions constraint. We show that up-to-date costs for wind and solar, that take into account recent capacity additions and technological learning, together with proper representation of balancing strategies make a fully decarbonised system based on those technologies cost-effective. Furthermore, we find that a transition path with more ambitious short-term $CO_2$ targets reduces the cumulative system cost and requires a smoother increase of the $CO_2$ price and more stable build rates. Our research includes the coupling with heating and transport sectors, which is absent in transition path analyses for the European power system[27–29], incorporates the notion of carbon budget to the analysis, and captures relevant weather-driven variability due to hourly and non-interrupted time stepping. Moreover, we use an open model, which ensures transparency and reproducibility of the results[35].

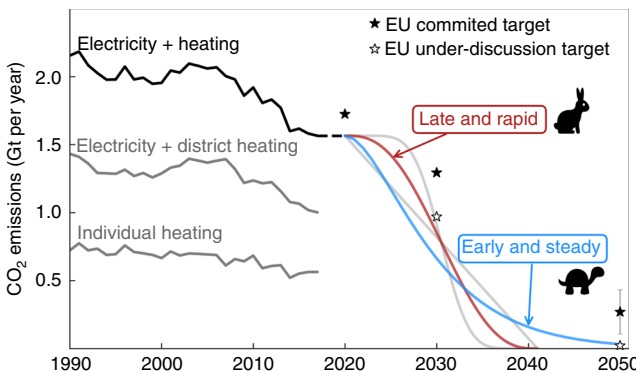

**Fig. 1 Historical $CO_2$ emissions from the European power system and heating supply in the residential and services sectors.** Data from EEA[8]. The various future transition paths shown in the figure have the same cumulative $CO_2$ emissions, which correspond to the remaining 21 $GtCO_2$ budget to avoid human-induced warming above 1.75 °C with a probability of >66%, assuming current sectoral distribution for Europe, and equity sharing principle among regions. Black stars indicate committed EU reduction targets, while white stars mark targets under discussion in 2020. See also Supplementary Fig. 1.

## Results

First, we investigate the consequences of following two alternative transition paths for the electricity and heating coupled system. The transport sector is added at the end of this section. The baseline analysis assumes that district heating penetration remains constant at present values, annual heat demand is constant throughout the transition paths, and power transmission

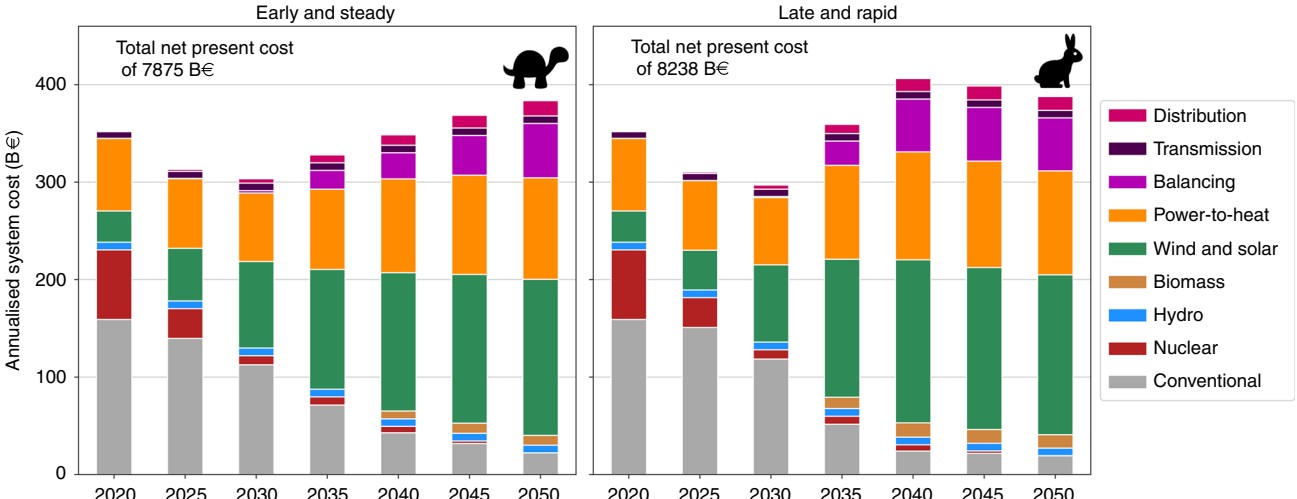

**Fig. 2 Annualised system cost for the European electricity and heating system throughout transition paths Early and Steady and Late and Rapid shown in Fig. 1.** Conventional includes costs associated with coal, lignite and gas power plants producing electricity as well as costs for fossil-fuelled boilers and CHP units. Power-to-heat includes costs associated with heat pumps and heat resistors. Balancing includes costs of electric batteries, $H_2$ storage and methanation.

capacities are expanded as planned in the TYNDP[36] up to 2030 and fixed after that year. The impacts of these assumptions are assessed later. The Early and Steady path represents a cautious approach in which significant emissions reductions are attained in the early years. In the Late and Rapid path, the low initial reduction targets quickly deplete the carbon budget, requiring a sharp reduction later. As in Aesop's fable "The Tortoise and the Hare", the tortoise wins the race by making steady progress, whereas following the hare and delaying climate action requires a late acceleration that will be more expensive.

**Cumulative costs and system configuration**. The two alternative paths arrive at a similar system configuration in 2050, Fig. 2. Towards the end of the period, under heavy $CO_2$ restriction, balancing technologies appear in the system. They include large storage capacities comprising electric batteries and hydrogen storage, and production of synthetic methane. Cumulative system cost for the Early and Steady path represents 7875 billion euros (B €), while the Late and Rapid path accounts for 8238 B€. It is worth remarking that the cumulative cost remains lower for the Early and Steady path provided that social discount rates <15% are assumed. In 2050, the cost per unit of delivered energy (including electricity and thermal energy) is ~59 €/MWh. The newly built conventional capacity for electricity generation is very modest in both cases, Fig. 3 and Supplementary Fig. 5. No new lignite, coal or nuclear capacity is installed. Thus, at the end of both paths, conventional technologies include only gas-fuelled power plants, CHP and boilers. Biomass contributes to balancing renewable power but plays a minor role.

Decarbonising the power system has proven to be cheaper than the heating sector[37]. Consequently, although $CO_2$ allowances differ, the electricity sector gets quickly decarbonised in both paths and more notable differences appear in new conventional heating capacities, Fig. 4. In both paths, yearly costs initially decrease as the power system takes advantage of the low costs of wind and solar. Removing the final emissions in heating causes total costs to rise again towards 2050. The main reason behind the higher cumulative system cost for the Late and Rapid strategy is that the earlier depletion of carbon budget forces it to reach zero emissions by 2040 when renewable generation and balancing technologies are more expensive than in 2050.

**Stranded assets**. Part of the already existing conventional capacities become stranded assets, in particular, coal, lignite, CCGT (which was heavily deployed in the early 2000s, Fig. 3) and gas boilers. As renewable capacities deploy, utilisation factors for conventional power plants decline and they do not recover their total expenditure via market revenues, Supplementary Figs. 11–14. Up to 2035, operational expenditure for gas-fuelled technologies are lower than market revenues so they are expected to remain in operation. Contrary to what was expected, the sum of expenditures not recovered via market revenues is similar for both paths. In the Late and Rapid path, the high $CO_2$ price resulting from the zero-emissions constraint, justify producing up to 220 TWh/a of synthetic methane already in 2040, Supplementary Fig. 10. This enables CCGT and gas boilers to keep operating allowing them to recover part of their capital expenditure, but the consequence is a higher cumulative system cost, as previously discussed. Stranded costs, that is the sum of expenditures not recovered via market revenues, represent ~12% of the total cumulative system cost in both paths. Although closing plants early might be seen as an unnecessary contribution to a higher cost of energy, it must be remarked that the early retirement of electricity infrastructure has been identified as one of the most cost-effective actions to reduce committed emissions and enable a 2 °C-compatible future evolution of global emissions[38].

**Transition smoothness**. Wind and solar PV supply most of the electricity demand in 2050, complemented by hydro and with a minor biomass contribution. Previously, most IAMs have emphasised the importance of bioenergy or carbon capture and storage and failed to identify the key role of solar PV due to their unrealistically high-cost assumptions for this technology, see refs. [10,34] and Supplementary Note 4.2. The paths described here require a massive deployment of wind and solar PV during the next 30 years. In the past, Germany and Italy have shown record installation rates for solar PV of 8 and 10 GW/a, Supplementary Fig. 4. Since those countries account for 16% and 10% of electricity demand in Europe, those rates would be equivalent to 50 and 100 GW/a at a European level. Decarbonising the electricity and heating sectors through the Early and Steady path requires similar installation rates, Fig. 3. Consequently, attaining higher build rates to also decarbonise transport and industry sectors seems challenging yet possible.

 3

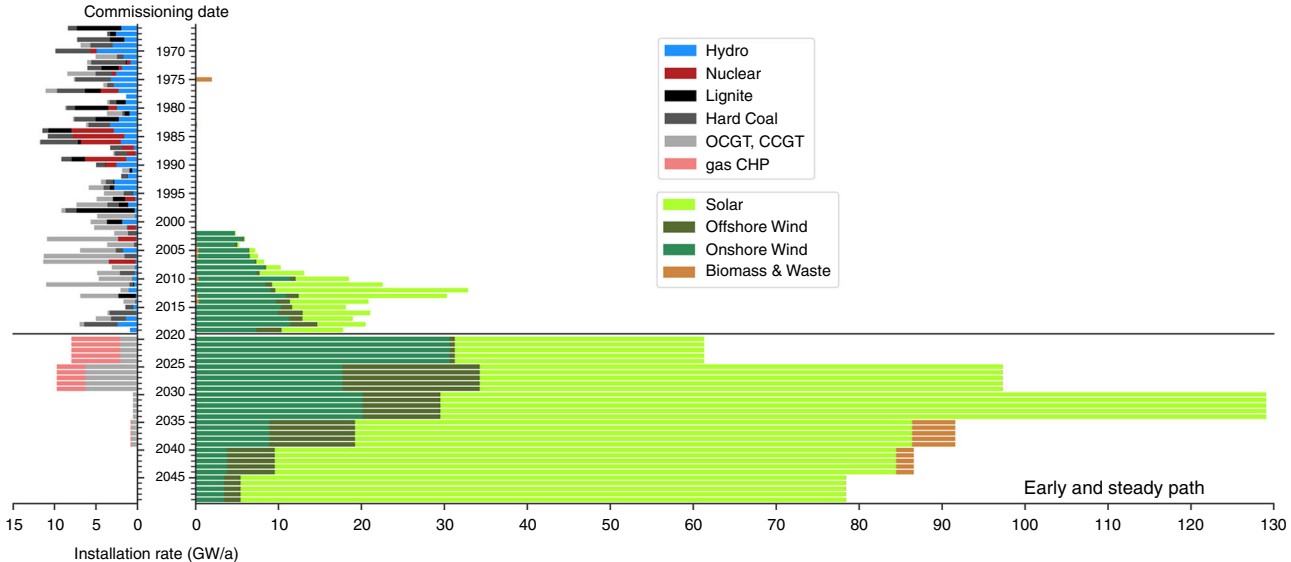

**Fig. 3 Age distribution of European power plants in operation and required annual installation throughout the Early and Steady path.** Historical data from refs. [53,67], see also Supplementary Figs. 5–10.

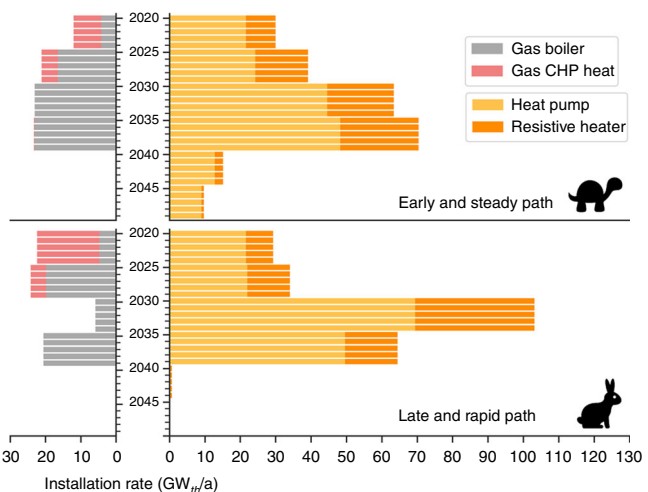

**Fig. 4 Required expansion of heating capacities in both paths.** Maximum heating capacities are shown for CHP plants.

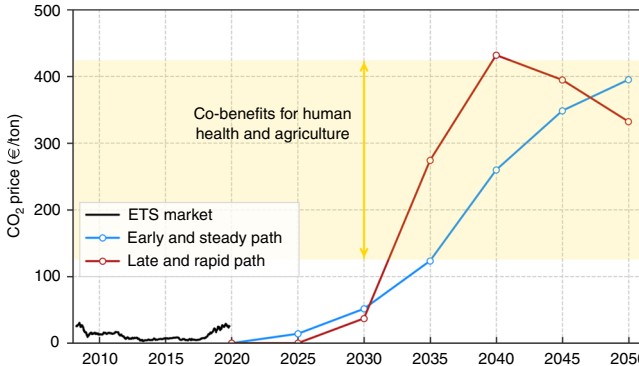

**Fig. 5 Historical evolution of $CO_2$ price in the EU Emissions Trading System[68] and required $CO_2$ price obtained from the model throughout transition paths shown in Fig. 1.** Co-benefits of reducing $CO_2$ emissions in Europe due to avoided premature mortality, reduced lost workdays and increased crop yields are estimated in the range of 125–425 €/ton $CO_2$[43].

During the past decade, several European countries have shown sudden increments in the annual build rate for solar PV, followed by equivalent decrements one or two years later, Supplementary Fig. 4. Italy, Germany, UK and Spain show clear peaks due to the combination of a fast cost decrease of the technology and unstable regulatory frameworks whose details are country-specific[39–41]. These peaks can have negative consequences for local businesses. The sudden shrinkage of annual build capacity might result in companies bankruptcy and lost jobs. The Early and Steady path requires a smoother evolution of build rates which could better accommodate the cultural, political, and social aspects of the transition[42], and Supplementary Fig. 15. The mild evolution could also facilitate reaching a stationary situation in which build rates offset decommissioning.

The required $CO_2$ price at every 5-years time step, Fig. 5, is an outcome of the model, i.e., it is the Lagrange/KKT multiplier associated with the maximum $CO_2$ constraint. The fact that results indicate zero $CO_2$ price in 2020 means that the constraint is not binding, that is, the cost of renewable technologies makes the system cost-effective without the constraint. As the $CO_2$ emissions are restricted, a higher $CO_2$ price is needed to remain below the $CO_2$ limit. Towards the end of the transition, $CO_2$ prices much higher than those historically attained in the ETS market are needed. The Early and Steady path requires a smoother evolution of $CO_2$ price, which might be preferred by investors. Two remarks should be made. First, reducing $CO_2$ emissions implies significant co-benefits in Europe associated with avoided premature mortality, reduced lost workdays and increased crop yields. Those cost benefits are estimated at 125–425 €/ton $CO_2$[43], which is similar to the required $CO_2$ prices at the end of the path. On top of that, economic benefits of mitigating climate change impacts have also been estimated in hundreds of €/ton $CO_2$. Second, $CO_2$ price is mainly an indicator of the price gap between polluting and clean technologies and several policies can be established to fill that gap. Among others, sector-specific $CO_2$ taxes[25], direct support for renewables that reduce investor risk, and consequently the cost of capital and LCOE of the technology[44], or regulatory frameworks that incentivise the required technologies such those promoting rooftop PV installations or ensuring the competitiveness of district heating systems.

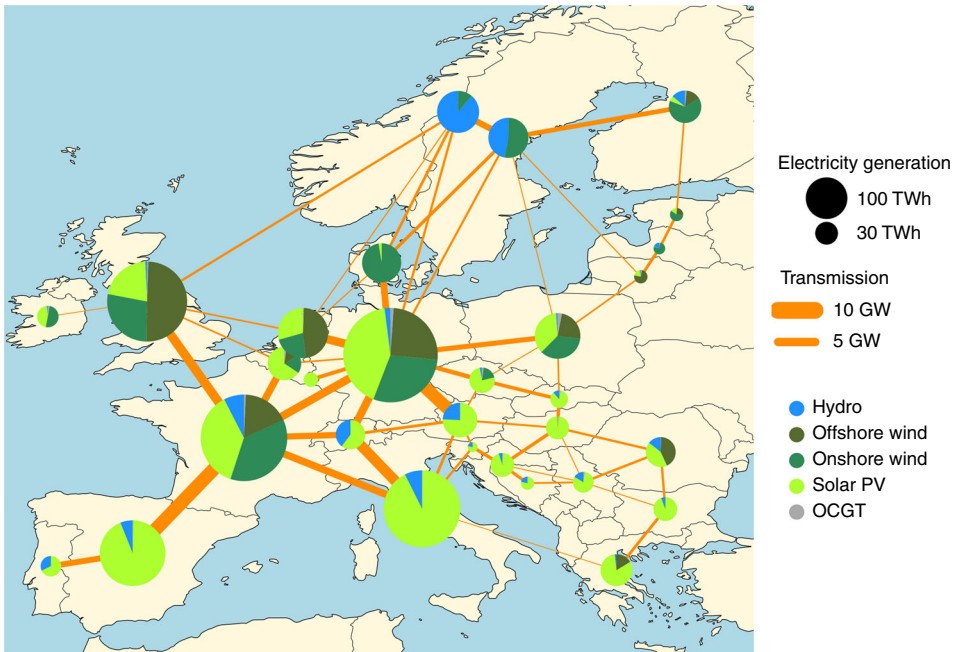

**Fig. 6 Electricity generation in 2050 in the Early and Steady path.** Evolution of the electricity mix throughout the transition and country-specific results are included in Supplementary Fig. 16.

**Country and hourly resolved results**. Figure 6 depicts the electricity mix at the end of the Early and Steady path. As expected, southern countries exploit solar resource while northern countries rely mostly on offshore and onshore wind. At every time step, the optimal renewable mix in every country depends on the local resources and the already existing capacities, see Supplementary Figs. 16 and 17. Nevertheless, the analysis of near-optimal solutions has recently shown that country-specific mixes can vary significantly while keeping the total system cost only slightly higher than the minimum[45].

Modelling an entire year with hourly resolution unveils the strong links between renewable generation technologies and balancing strategies. For countries and years in which large solar PV capacities are deployed, it is also cost-effective to install large battery capacities to smooth the strong daily solar generation pattern. Conversely, onshore and offshore wind capacities require hydrogen storage and reinforced interconnections to balance wind synoptic fluctuations[13,17,19]. This can also be appreciated by looking at the dominant dispatch frequencies of the Europe-aggregated time series in 2050, Fig. 7 and Supplementary Fig. 18.

IAMs and partial equilibrium models with similar spatial resolution have also been used to investigate the sector-coupled decarbonisation of Europe[1,10,46]. However, those models typically use a much lower time resolution, e.g., using a few time slices to represent a full year[29,46–49] or considering the residual load duration curve[10,50], and some IAMs assume very high integration costs for renewables[51]. The hourly and non-interrupted time stepping in our model reveals several effects that are critical to the operation of highly renewable systems. First, solar and wind power generation is variable but correlated. The grid can effectively contribute to its smoothing by regional integration and storage technologies with different dispatch frequencies required to balance solar and wind fluctuations, Fig. 7. Second, long-term storage plays a key role in balancing seasonal variation and ease the system operation during cold spells, i.e., a cold week with low wind and solar generation[21].

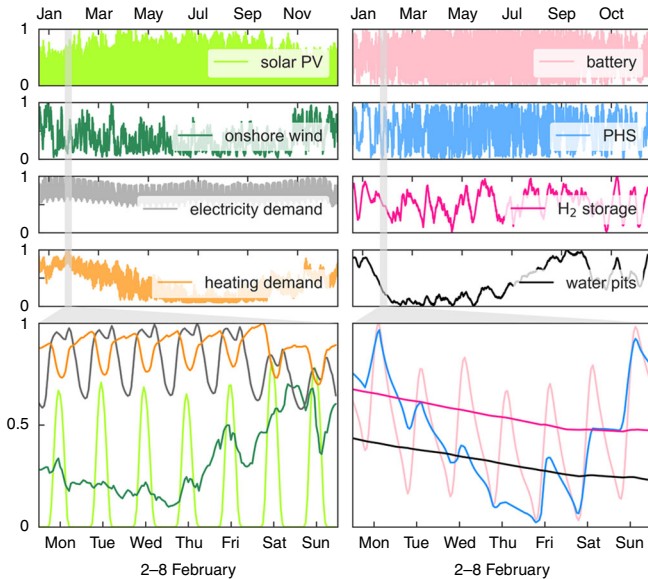

**Fig. 7 Time series for the Europe-aggregated demand, generation and storage technologies dispatch for the Early and Steady path in 2050.** The bottom figures depicts the system operation throughout one of the most critical weeks of the year (comprising high heating demand, low wind and solar generation). Hydrogen storage discharges and fuel cells help to cover the electricity deficit, hot water pits in district heating systems discharge stored thermal energy to supply heat demand.

**Results robust under different scenarios.** In Nordic countries, district heating (DH) has proven to be useful to decarbonise the heating sector, Supplementary Fig. 2. It allows lower cost large-scale technologies such as heat pumps and CHP units, enables a faster conversion because it is easier to substitute one central heating unit than a myriad of individual domestic systems, and facilitates long-term thermal energy storage, via cheap large water pits, Fig. 7, that help to balance the large seasonal variation of heating demand, Supplementary Fig. 24. So far, we have assumed

**Table 1 Cumulative system costs (B€) for additional analyses.**

| Analysis | Early and Steady path | Late and Rapid path | Difference | Change relative to Baseline (Early and Steady) (%) |
|---|---|---|---|---|
| Baseline | 7875 | 8238 | 363 | |
| District heating expansion | 7688 | 8003 | 315 | −187 (−2.4) |
| Space heat savings due to building renovation | 6989 | 7319 | 330 | −886 (−11.3) |
| Transmission expansion after 2030 | 7771 | 8081 | 310 | −104 (−1.3) |
| Including road and rail transport | 8303 | 8753 | 450 | +428 (+5.4) |

that DH penetration remains constant at 2015 values. When DH is assumed to expand linearly so that in 2050 it supplies the entire urban heating demand in every country, cumulative system cost for the Early and Steady path reduces by 2.4%. This roughly offsets the cost of extending and maintaining the DH networks and avoids the additional expansion of gas distribution networks, Supplementary Note 4.5.

We now look at the impact of efficiency measurements by modifying the constant heat demand assumption. When a 2% reduction of space heating demand per year is assumed due to renovations of the building stock, while demand for hot water is kept constant and rebound effects are neglected, cumulative system cost decreases by 11.3%, significantly offsetting costs of renovations, Supplementary Note 4.6.

When the model is allowed to optimise transmission capacities after 2030, together with the generation and storage assets, the optimal configuration at the end of the paths includes a transmission volume approximately three times higher than that of 2030. The reinforced interconnections contribute to the spatial smoothing of wind fluctuations, increasing the optimal onshore and offshore wind capacities at the end of the path. The required energy capacity for hydrogen storage is reduced due to the contribution of interconnections to balancing wind generation. Although the cumulative system cost is 1.3% lower, it is unclear to what extent it compensates the social acceptance issues associated with extending transmission capacities.

Neither of the paths installs new nuclear capacity. This technology is only part of the optimal system in 2050 when nuclear costs are lower by 15% compared to the reference cost and no transmission capacity expansion is allowed. In all the previous scenarios, the difference in cumulative system cost for the Early and Steady and the Late and Rapid path is roughly the same, Table 1.

**Adding the transport sector**. Finally, both paths are re-run including the coupling of road and rail transport. The number of Battery Electric Vehicles (BEVs) is not an outcome of the optimisation but an exogenous input parameter. This assumes that people's decision to shifts to BEVs is mainly dictated by their mobility needs and not by the optimal operation of the energy system. The cost of BEVs and their batteries are not included in the results. For every time step, the percentage of road and rail transport that is electrified is assumed to follow the path of $CO_2$ emissions reduction in the electricity and heating sectors. In this way, emissions in transport sink roughly parallel to those of heating and electricity sectors. Road and rail transport is modelled as a lumped demand in every country. The details of the model for this sector are described in Supplementary Note 3.5. At every time step, half of the passenger car BEVs present in the model are assumed to allow demand-side management and a quarter of the available BEVs are assumed to provide vehicle-to-grid (V2G) services. The possible use of hydrogen in the transport sector is not considered.

For the Early and Steady path, cumulative system cost increase by 5.4%. The system cost increase was expected, since, when fully electrified, road and rail transport increase electricity demand by 1102 TWh$_{el}$/a. However, the evolution of LCOE remains similar throughout the transition, Supplementary Figs. 6 and 20. The additional flexibility provided by BEVs reduces the need for stationary batteries and incentivises a higher solar PV penetration, as previously observed[19,21]. The impacts of the percentage of BEVs allowing smart charging and V2G services were analysed in detail in Brown et al.[21] where it is shown that the initial 25% of vehicles doing V2G captures the highest cost reductions.

**Discussion**

In this section, we briefly compare our results with other relevant decarbonisation pathways for Europe and indicate the main limitations of this study.

The analysis accompanying the EU *Clean Planet for All* strategy[1] comprises 8 scenarios, three of which are compatible with limiting temperature increase at the end of the century to 1.5 °C. All of them include a nuclear capacity >85 GW in 2050. Most probably this is a result of the lower cost assumed for nuclear in ref. [1]. Scenario 1.5Life in ref. [1] assumes significant changes in lifestyle and consumer choices, while Scenario 1.5Tech relies on bioenergy with carbon capture and storage (BECCS). In ENTSO-E scenario report[36], biomass accounts for >30% of the electricity mix in 2050. Using cost-optimisation we have shown that a decarbonised European electricity mix based mainly on wind and solar is cost-effective. It can also avoid the concerns associated with nuclear, biomass and BECCS. A proper evaluation of feasibility requires a multidimensional approach which on top of the land availability, technological and economical aspects considered here, includes also social acceptance, institutions and politics. Although that evaluation is out of the scope of this work, the gradual transition described in the Early and Steady path could potentially be beneficial when those aspects are taken into consideration.

A recent analysis of the globally cost-effective emission pathways for the emissions cap in the EU ETS showed that increasing the linear reduction factor for 2021–2030 from the current value of 2.2 to 4% is cost-effective[52]. This is supported by the increase in renewable penetration and efficiency targets for 2030 and the coal phase-out plans of several European countries. For the ETS sectors, failing to reduce emissions in the next decade would require a drastic reduction after 2030 that implies higher cumulative costs[52]. The results in this paper, which include also non-ETS sectors such as transport and domestic heating supply, support this recommendation.

The database of existing power plants was described and validated in a separate publication[53]. The power system model PyPSA-Eur including load, generation and a detail transmission network, was validated in Hörsch et al.[54], while the interplay of generation and network with regards historical curtailment levels was examined in Frysztacki et al.[55]. Data on the existing heating was taken from ref. [56].

Our model uses hourly resolution, but as renewable penetration increases, adaptation will also be required to ensure system stability at shorter time scales. Several strategies are being developed and implemented to ensure sufficient power system inertia and the provision of reserve requirements and ancillary services[15,57]. Synchronous compensators to provide reactive power and inertia are already used in Denmark[58] and conventional power plants can be retrofitted to become such synchronous compensators. Grid-forming inverters in batteries and non-synchronous generators can regulate the system frequency and voltage[57,59]. Solar and wind generation can contribute to downward regulation by curtailing and to upward regulation when operating at reduced capacity as well as storage and demand response from new electrified loads like electric vehicles and heat pumps[57]. The existing literature[15,57] and historical field experience do not indicate any major limitation to ensure the feasibility of highly renewable power system at short time scales.

This study uses one single year of weather data. The system cost for a highly decarbonised European power system was found to be robust to different weather years[60], but more analysis is needed on the impact of inter-annual weather variability for the sector-coupled energy system. Climate change will have a twofold impact. On the generation side, correlation lengths for wind energy are predicted to increase in Europe, reducing the efficacy of transmission grids for balancing[61]. Minor changes in solar generation[62], and significant variations on the hydro inflow seasonal patterns are expected[61]. On the demand side, the increase in cooling demand in southern European countries, and more relevant, the reduction of heating demand in northern countries are expected to reduce the system cost[63].

Low social acceptance for onshore wind and utility-scale solar may limit expansion of these technologies in some countries. Reducing onshore wind installable potentials was shown in Schlachtberger et al.[60] to cause a larger expansion of offshore wind and have a limited impact on total system costs. In our model, hydrogen is assumed to be produced and consumed within the same country and transport of hydrogen is not included. The possible future retrofitting of existing nuclear power plants or the deployment of coal power plants with carbon capture and storage (CCS) are not included in the model. The aforementioned limitations could impact the model results but they are not expected to modify the main conclusions obtained in this study. Finally, this study focuses on the European energy system because the European Union has a shared decarbonisation strategy and a common commitment via the Paris agreement. Moreover, the power systems of member states are already interconnected. Neglecting the interconnection of Europe with other regions and their mutual interdependence is a limitation of the analysis.

In conclusions, when comparing alternative transition paths for the European energy system with the same carbon budget, we find that a transition including an early and steady $CO_2$ reduction is consistently ~350 B€UR cheaper than a path where low targets in the initial period demand a sharper reduction later. Our results support the proposal to increase the ambition in the EU $CO_2$ reduction target for 2030 under discussion in 2020[4]. We found that up-to-date costs for wind and solar and the inclusion of highly resolved time series for balancing allows a fully decarbonised system relying on those technologies together with hydropower and minor contribution from biomass. The required renewable build rates to decarbonise the electricity and heating sectors correspond to the highest historical values, making the transition challenging yet possible. We have shown that early action not only allows room for decision-making later but it also pays off.

## Methods

The system configuration is optimised by minimising annualised system cost in every time step (one every 5 years), under the global $CO_2$ emissions cap imposed by the transition path under analysis (Fig. 1). This can be considered a myopic approach since the optimisation has no information about the future. The cumulative $CO_2$ emissions for the transition paths is equal to a carbon budget of 21 $GtCO_2$ when only the electricity and heating sectors are included. It represents 33 $GtCO_2$ when the transport sector is included. In every time step, generation, storage and transmission capacities in every country are optimised assuming perfect competition and foresight as well as a long-term market equilibrium. Besides the global $CO_2$ emission cap, other constraints such as the demand-supply balance at every node, and capacity limitations are imposed to ensure the feasibility of the solution, see Supplementary Note 2.

We use a one-node-per-country network, including 30 countries corresponding to the 28 European Union member states as of 2018 excluding Malta and Cyprus but including Norway, Switzerland, Bosnia-Herzegovina and Serbia, see Fig. 6. Countries are connected by High Voltage Direct Current (HVDC) links whose capacities can be expanded if it is cost-effective. Figure 8 provides an overview of

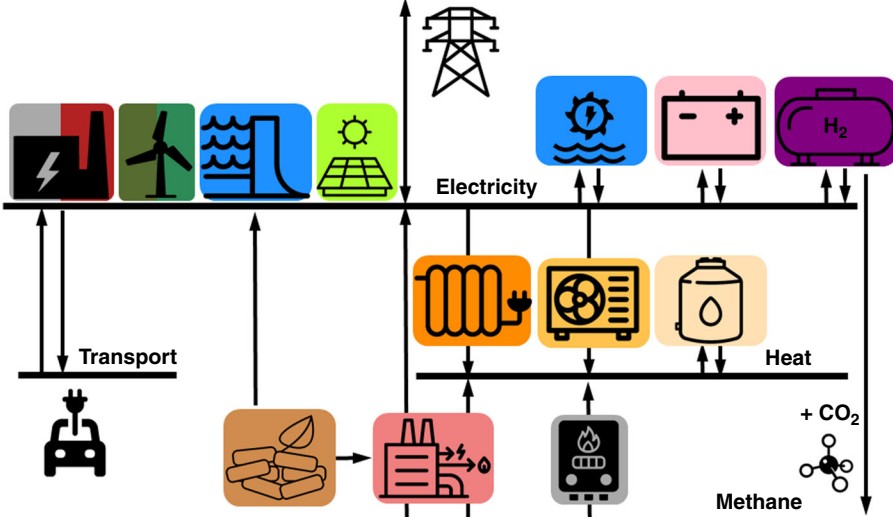

**Fig. 8 Model diagram representing the main generation and storage technologies in every country.** Electricity can be produced with conventional power plants (nuclear, coal, lignite and gas), reservoir and run-of-river hydro, as well as solar PV, onshore and offshore wind. It can be stored in pumped hydro storage, battery and hydrogen storage. Heating can be supplied via heat resistors, heat pumps and gas boilers. It can be stored in thermal energy storage. Combined heat and power (CHP) can use methane and biomass. Electrolytic hydrogen can be combined with direct air capture $CO_2$ to produce synthetic methane. When the transport sector is included, half of the battery electric vehicles allow smart charging and a quarter provide vehicle-to-grid services.

the technologies included in the model. In the power sector, electricity can be supplied by onshore and offshore wind, solar photovoltaics (PV), hydroelectricity, Open Cycle Gas Turbines (OCGT), Combined Cycle Gas Turbines (CCGT), Coal, Lignite, and Nuclear power plants and Combined Heat and Power (CHP) units using gas, coal or biomass. Electricity can be stored using Pumped Hydro Storage (PHS), stationary electric batteries and hydrogen storage. Hydrogen is produced via electrolysers and converted back into electricity using fuel cells. Methane can be produced by combining Direct Air Capture (DAC) $CO_2$ and electrolytic-$H_2$ in the Sabatier reaction. Heating demand is split into urban heating demand, corresponding to regions whose population density allows district heating, and rural heating demand where only individual solutions are allowed. Heating can be supplied via large-scale heat pumps, heat resistors, gas boilers, solar collectors and CHP units for urban regions, while only individual heat pumps, electric boilers and gas boilers can be used in rural areas. Thermal energy storage can be installed both in district heating networks and in individual homes. A detailed description of all the sectors is provided in Supplementary Note 3.

Costs assumed for the different technologies depend on time (Supplementary Note 4) but not on the cumulative installed capacity since we assume that they will be influenced by the predicted global installation rates and learning curves. The financial discount rate applied to annualise costs is equal to 7% for every technology and country. Although it can be strongly impacted by the maturity of a technology, including the country-specific experience with the technology, and the credit rating of a country[64], we assumed European countries to be similar enough to use a constant discount rate. For decentral solutions, such as rooftop PV or small water tanks, a discount rate equal to 4% is considered based on the assumption that individuals have lower expectations for return on capital[65]. The already installed capacities, i.e., existing capacities in 2020 or capacities installed in a previous year whose lifetime has not concluded, are exogenously included in the model. For every time step, the total system cost includes annualised and running cost for newly installed assets and for exogenously fixed capacities. For those fossil fuel generators that were installed in a previous year and are not used due to the stringent $CO_2$ emissions constraint, their annualised costs are included in the total system cost (see Fig. 2 in the Results section) as long as the end of their assumed technical lifetime is not reached.

To estimate the cumulative cost of every transition path, the annualised cost for all years are added up assuming a social discount rate of 2%. This rate represents how society considers investments in far-future years with investments in the present, and is chosen by comparison with the average growth rate of 1.6% over the past 20 years in the European Union. The $CO_2$ price is not an input to the model, but a result that is obtained via the Lagrange/Karush-Kuhn-Tucker multiplier associated with the global $CO_2$ constraint, see Supplementary Note 2.

## Data availability
The datasets used as input as well as the data generated by the model are available in a public repository 10.5281/zenodo.4010643.

## Code availability
The model is implemented in the open-source framework Python for Power System Analysis (PyPSA)[66]. The model instance used in this paper can be retrieved from the open repository 10.5281/zenodo.4014807.

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

## Acknowledgements

M.V., K.Z., G.B.A. and M.G. are fully or partially funded by the RE-INVEST project, which is supported by the Innovation Fund Denmark under grant number 6154-00022B. T.B. acknowledges funding from the Helmholtz Association under grant no. VH-NG-1352. The responsibility for the contents lies solely with the authors.

## Author contributions

M.V. designed the analysis, drafted the manuscript and contributed to the data acquisition, analysis and interpretation of data. K.Z. contributed to the data acquisition, modelling, analysis and interpretation of data. T.B., G.B.A. and M.G. contributed to the initial idea and made substantial revisions of the manuscript.

## Competing interests

The authors declare no competing interests.
