## [Peer Review File · Nature Communications]

Reviewer #1 (Remarks to the Author):

This is a timely and well-written paper and I think it suitable to the broad readership of Nature Communications though it needs to be revised. The early and steady approach being less costly and challenging makes intuitive sense, a more constrained optimisation will not be cheaper. The key strength of this work is that is far more resolved and up to date than most IAM studies. It is useful to see this explored in a European context though I feel it could be better unpacked why this is the case.

Given that electricity will play a key role in making either of these scenarios a reality, what is the impact of long-term variations of the variable renewables that underpin this decarbonisation and is it sufficiently robust? This was unclear in the supplementary material and will be key, this is aside from the impact of climate change on these resources.

A power system based on so much zero-marginal cost variable renewables will fundamentally change how it is operated with implications for power system inertia, reserve requirements and ancillary services market design. I feel that this is not discussed or alluded to sufficiently in the text. IAMs do have their weaknesses, but a strength is that they provide an integrated global approach, this implies a weakness in modelling a single region alone that could be mentioned or further explored in the text.

These are some heroic assumptions for EV batteries power sector, 50% of the fleet being available for battery cycling is a lot, this merits more explanation in the main body. Generally speaking, I think the description of the transport sector modelling in the paper isn't sufficiently detailed to fully grasp how it is modelled. "For every time step, the electrification of transport is assumed to be equal to the CO₂ emissions reduction relative to 2020." is ambiguous, I understand the decarbonisation must follow a similar track but this sounds like electricity and emissions have been equated. There has also been some interesting recent literature regarding the electrification of roads with catenary lines which would reduce battery requirements in electric vehicles.

There is solid policy context at the beginning but this is lost at the end when conclusions are drawn, it would be useful for the reader that this be made more punchy and tied in with this reality.

Well done again on an interesting and well-written paper with such extensive modelling work.

Reviewer #2 (Remarks to the Author):

The paper addresses the most cost-effective implementation of energy system transformation in Europe, and thus a very relevant topic for a wide readership. It is based on an innovative, transparent and detailed modelling approach, which is well explained, as are the data used and the assumptions made. However, the (informed) reader is required to keep jumping back and forth between sections 2 and 4, as the results cannot be fully understood without prior knowledge of the method. The same applies to much information in the Supplementary Material. Limitations of the method and data assumptions are insufficiently mentioned and discussed to some degree, as explained in detail below. The paper is written well and understandably. The paper has a significant methodological added value in that the conclusions described are very well supported quantitatively. One of the novelty values mentioned is to show that a European energy supply, which is mainly based on wind and solar energy, is possible and cheaper than the alternatives. It is also shown that an early start to the transformation is more favourable than a later one. Both aspects are already taken up in the cited literature, but are not proven on the basis of such a comprehensive

methodology and combined. The proof that a transformation can be realized with historically already observed growth rates has not yet been provided. However, it is not shown here for a complete sector-coupled energy system due to the restriction to electricity and heat. Still, it is a significant step forward compared to the existing literature.

Model configuration and data

- Is the climate change impact on the energy system (heating and cooling demand, but also beyond) considered? If not, please discuss?
- Can the model invest in the refurbishment of power plants at reduced costs compared to a new installation? This might have an impact especially on the role of nuclear, possible also other fossils and renewables. Can existing coal power plants be enhanced to include CCS? If not, please discuss both aspects.
- Please add a short paragraph on model validation
- The role of hydrogen does not become completely clear to me. Are production and re-electrification taking place at the same (storage) site? Is the heat from fuel cells used? Do you consider transport of hydrogen and heat that might be related? If not, please discuss.
- Are system reliability and security explicitly included in the model? Please discuss.

Additional scenarios

- You highlight the cost reductions that can be achieved by DH extension, building refurbishment and additional power transmission. In order to better classify the achievable cost savings, the addition of percentage values in the text would be helpful
- Social acceptance might not only be an issue regarding grid expansion. I propose to separate the statement from the corresponding paragraph and make it more general
- Why is the transport sector not included in the main analysis? What is the impact of (flexibly charged) electric driving on the power and heat supply, stationary battery capacity and so on? What is the impact on RE installation rates? Please discuss.

Reviewer #1:

Dear reviewer #1,

We want to thank you for the time that you devoted to review our manuscript and to make suggestions to improve it. All your comments have been carefully considered and your suggested modifications have been implemented. A new version of the manuscript where all the changes are marked in red is also provided. Please, find detailed answers to your comments below.

Reviewer #1 (Remarks to the Author):

This is a timely and well-written paper and I think it suitable to the broad readership of Nature Communications though it needs to be revised. The early and steady approach being less costly and challenging makes intuitive sense, a more constrained optimisation will not be cheaper. The key strength of this work is that is far more resolved and up to date than most IAM studies. It is useful to see this explored in a European context though I feel it could be better unpacked why this is the case.

Given that electricity will play a key role in making either of these scenarios a reality, what is the impact of long-term variations of the variable renewables that underpin this decarbonisation and is it sufficiently robust? This was unclear in the supplementary material and will be key, this is aside from the impact of climate change on these resources.

We have added a paragraph to the Discussion section in which we provide additional information on the impact of long-term variations of variable renewable generation and the expected impact of climate change on highly renewable energy systems. While inter-annual variations of renewables in the power system have been addressed in the literature, more analysis is needed for sector-coupled systems which are particularly impacted on the demand side by inter-annual changes in space heating demand.

A power system based on so much zero-marginal cost variable renewables will fundamentally change how it is operated with implications for power system inertia, reserve requirements and ancillary services market design. I feel that this is not discussed or alluded to sufficiently in the text.

We have added a paragraph discussing the feasibility of a highly renewable power system with regards to inertia, reserve requirements and ancillary services.

IAMs do have their weaknesses, but a strength is that they provide an integrated global approach, this implies a weakness in modelling a single region alone that could be mentioned or further explored in the text.

This limitation has also been added to the Discussion section.

These are some heroic assumptions for EV batteries power sector, 50% of the fleet being available for battery cycling is a lot, this merits more explanation in the main body. Generally

speaking, I think the description of the transport sector modelling in the paper isn't sufficiently detailed to fully grasp how it is modelled. "For every time step, the electrification of transport is assumed to be equal to the CO2 emissions reduction relative to 2020." is ambiguous, I understand the decarbonisation must follow a similar track but this sounds like electricity and emissions have been equated. There has also been some interesting recent literature regarding the electrification of roads with catenary lines which would reduce battery requirements in electric vehicles.

We have improved the description of the transport sector modelling in the main text. Unfortunately, due to the limit space available, we have had to keep the details in the Supplementary Materials. We have also added a reference where the impacts of different availability of EV batteries, smart-charging batteries, and vehicle-to-grid is investigated in further detail. Note that we only use demand response flexibility from passenger BEVs, thus leaving open the possibility for catenary lines to be used by heavy transport. Our assumptions are that 50% of passenger BEVs can shift their charging in time (not implying extra cycling) while only 25% do V2G (implying possible extra cycling). Note that modern car batteries have significantly longer lifetimes given cycling than found in some historical literature, so that battery degradation should be low (see e.g. <https://doi.org/10.1016/j.joule.2019.06.002>). Low levels of deep cycling with V2G were seen in our previous publication <https://doi.org/10.1016/j.energy.2018.06.222>.

There is solid policy context at the beginning, but this is lost at the end when conclusions are drawn, it would be useful for the reader that this be made more punchy and tied in with this reality.

We have added a paragraph in the discussion section relating our results with the recent proposal of increasing the linear reduction factor of the ETS cap in the 2020-2030 period and how our results support this recommendation.

Furthermore, we have added an explicit reference to the ongoing discussion on increasing CO₂ reduction ambition for 2030 and how our results support this proposal.

Well done again on an interesting and well-written paper with such extensive modelling work.

We would also like to mention that, during the revision process, we discovered a small bug in the code that affects the calculated cumulative system cost. However, the main tendencies for the Early and Steady and the Late and Rapid paths remain the same and so does the cumulative cost difference and the other results in the paper. We have updated Figure 3 and Table 1 with the corrected values.

Reviewer #2

Dear reviewer #2,

We want to thank the time that you devoted to review our manuscript and to make suggestions to improve it. All your comments have been carefully considered and suggested modifications have been implemented. A new version of the manuscript where all the changes are marked is also provided. Please, find detailed answers to your comments below.

Reviewer #2 (Remarks to the Author):

The paper addresses the most cost-effective implementation of energy system transformation in Europe, and thus a very relevant topic for a wide readership. It is based on an innovative, transparent and detailed modelling approach, which is well explained, as are the data used and the assumptions made. However, the (informed) reader is required to keep jumping back and forth between sections 2 and 4, as the results cannot be fully understood without prior knowledge of the method. The same applies to much information in the Supplementary Material. Limitations of the method and data assumptions are insufficiently mentioned and discussed to some degree, as explained in detail below. The paper is written well and understandably. The paper has a significant methodological added value in that the conclusions described are very well supported quantitatively. One of the novelty values mentioned is to show that a European energy supply, which is mainly based on wind and solar energy, is possible and cheaper than the alternatives. It is also shown that an early start to the transformation is more favourable than a later one. Both aspects are already taken up in the cited literature, but are not proven on the basis of such a comprehensive methodology and combined. The proof that a transformation can be realized with historically already observed growth rates has not yet been provided. However, it is not shown here for a complete sector-coupled energy system due to the restriction to electricity and heat. Still, it is a significant step forward compared to the existing literature.

We have moved the Methods section prior to the Results section. We believe this facilitates the reading and understanding of the results in the paper. Unfortunately, due to the limit space available, we had to restrict the details on the description of the method and refer to the Supplementary Materials.

Following your recommendation, we have added some additional text in the Discussion section in which the main limitations are briefly discussed. The limitations of this study are discussed in the last three paragraphs of the Discussion section.

Model configuration and data

- Is the climate change impact on the energy system (heating and cooling demand, but also beyond) considered? If not, please discuss?

No, it is not included. We have added a paragraph in which we discuss the main impact of climate change on the energy system.

- Can the model invest in the refurbishment of power plants at reduced costs compared to a new installation? This might have an impact especially on the role of nuclear, possible also other fossils and renewables. Can existing coal power plants be enhanced to include CCS? If not, please discuss both aspects.

The model cannot invest in retrofitting power plants at reduced costs. Coal power plants with CCS are not included in the model, since many countries have set dates to end coal generation and it has been shown in other studies that coal with CCS is not cost-competitive because of its high capital costs and low utilisation rates (see e.g. <https://doi.org/10.1016/j.energy.2012.06.002>).

This has been mentioned as a model limitation in the Discussion section.

- Please add a short paragraph on model validation

The following paragraph on validation was added to the discussion section.

The database of existing power plants was described and validated in a separate publication [40]. The power system model PyPSA-Eur including load, generation and a detail transmission network, was validated in [58], while the interplay of generation and network with regards historical curtailment levels was examined in [59]. Data on the existing heating was taken from [60].

- The role of hydrogen does not become completely clear to me. Are production and re-electrification taking place at the same (storage) site? Is the heat from fuel cells used? Do you consider transport of hydrogen and heat that might be related? If not, please discuss.

Yes, production and consumption are assumed to take place at the same (storage) site. Pipeline transport would be possible and is under discussion in Europe, but large-scale international hydrogen transport is not yet done today. A full consideration of the hydrogen demand in industry (for ammonia and steel production) would be needed to assess pipeline requirements.

The heat from fuel cells is not used. We do not consider the transport of hydrogen as a possibility.

This has been mentioned as a limitation in the Discussion section.

- Are system reliability and security explicitly included in the model? Please discuss.

As mentioned in the Supplementary Note 3.1

For the transmission line capacities F_i , a safety margin of 33% of the installed capacity is reserved to satisfy $n-1$ requirements, following standard practice in the literature

We have added a paragraph discussing the feasibility of a highly renewable power system with regards to inertia, reserve requirements and ancillary services.

Additional scenarios

- You highlight the cost reductions that can be achieved by DH extension, building refurbishment and additional power transmission. In order to better classify the achievable cost savings, the addition of percentage values in the text would be helpful

Thanks for the suggestion, we have added percentage values to the text and Table 1.

- Social acceptance might not only be an issue regarding grid expansion. I propose to separate the statement from the corresponding paragraph and make it more general

It is true that social acceptance might also be an issue with other elements of the transition. However, we believe that its impact on the cost of extending transmission capacity is particularly relevant. In the past, the real cost of extending transmission capacities has been sometimes much higher than initially estimated due to delays caused by social acceptance issues. Hence, we prefer to keep this sentence the way it is now to highlight the small cost difference in the two alternative scenarios (w/w-o transmission expansion after 2030) and opposed it to the possible underestimation of “social acceptance costs”. Social acceptance as part of more general feasibility is mentioned briefly in the Discussion.

- Why is the transport sector not included in the main analysis? What is the impact of (flexibly charged) electric driving on the power and heat supply, stationary battery capacity and so on? What is the impact on RE installation rates? Please discuss.

The road and rail transport are not included in the analysis discussed in the first place, but it is included in the final part of the Results section. The main reason is that the number of electric vehicles in the grid is not an outcome of the optimization but a parameter exogenously fixed. This is because we do not think the number of EV at every time step will be a result of the role that they can play in the energy system but more a result based on people’s decision on how to satisfy their mobility needs, and how that is impacted by vehicle cost developments.

Hence, we decided to start by limiting the analysis to the electricity and heating sectors, in which the newly installed capacities are determined by the optimiser and no major exogenous decision is impacting the results. Then, after that analysis, we included the time evolution of the number of EVs in the system using a heuristic rule (“rate of electrification in transport equal to the reduction of CO2 emissions in electricity and heating sectors”) and investigate its impact on the system.

For the impact on stationary battery capacity, the main text reads *“The additional flexibility provided by EVs reduces the need for stationary batteries and incentivises a higher solar PV penetration, as previously observed [19, 21]”*.

Due to the limited available space, we did not discuss the impact on RE installation rates, since this was discussed in a previous publication (<https://doi.org/10.1016/j.energy.2018.06.222>). We have added Figure 22 to the Supplementary materials with the evolution of installed capacities when the transport sector is included.

We would also like to mention that, during the revision process, we discovered a small bug in the code that affects the calculated cumulative system cost. However, the main tendencies for the Early and Steady and the Late and Rapid paths remain the same and so does the cumulative cost difference and the other results in the paper. We have updated Figure 3 and Table 1 with the corrected values.

Reviewer #1 (Remarks to the Author):

The authors have amended the paper in line with my comments and I am satisfied with how they have done so. The important effects of inter-annual variability of weather-driven renewable resources in addition specificities of power and transport modelling have been incorporated. When combined with more specific and consistent policy context I believe the paper is suited for publication with Nature Communications.

Reviewer #2 (Remarks to the Author):

Dear authors,
thank you very much for the extensive and conscientious revision of the manuscript and the convincing response to my comments. I have no further comments. Congratulations on this very successful work.